# Differences in COVID-19 testing perceptions among caregivers of children with medical complexity by rurality

Kristina Devi Singh-Verdeflor[1]*, Michelle M. Kelly[1], Gregory P. DeMuri[1], Gemma Warner[1], Sabrina M. Butteris[1], Mary L. Ehlenbach[1], Barbara Katz[2], Joseph A. McBride[1,3], Shawn Koval[4], Ryan J. Coller[1]

1 Department of Pediatrics, University of Wisconsin School of Medicine and Public Health, Madison, Wisconsin, United States of America, 2 Family Voices of Wisconsin, Madison, Wisconsin, United States of America, 3 Department of Medicine, University of Wisconsin School of Medicine and Public Health, Madison, Wisconsin, United States of America, 4 University of Wisconsin Health, Office of Population Health, Madison, Wisconsin, United States of America

* ksinghverdef@wisc.edu

## Abstract

### Background

COVID-19 testing safeguards the health of children with medical complexity (CMC) through several key mechanisms, such as the implementation of clinical action plans and COVID-19-directed therapies. However, testing utility is limited by barriers to access and perceptions surrounding use. This study investigated associations between rurality and COVID-19 testing access, intent, motivators, and concerns for caregivers of CMC.

### Methods

We conducted a cross-sectional survey (April – June 2022) of English- and Spanish-speaking caregivers of children with at least one complex chronic condition between ages 5–17 at an academic medical center in the Midwestern USA. Rurality was dichotomized using Rural-Urban Commuting Area codes. Outcomes represented COVID-19 testing access, intent, motivators, and concerns. Covariates included demographic and clinical characteristics. Unadjusted and adjusted logistic regression analyses examined associations between rurality and each outcome.

### Results

Among 1,432 responses (response rate 49%), 359 (25%) were classified as rural. Respondents had varied education, income, and insurance levels. In the multivariable models, rural and urban caregivers reported similarly high testing access, but rural caregivers had significantly less testing intent (adjusted Odds Ratio [95% CI]: 0.53, [0.40, 0.71]). Notably, rural caregivers were significantly more likely to indicate "It will be difficult to get needed healthcare if my child has it" (2.49 [1.19, 5.18]).

**Data availability statement:** Deidentified study data and materials are available through the NIH RADx-UP Data Hub at https://radxdatahub.nih.gov/study/156. DOI: 10.60773/61bs-2j03

**Funding:** RC was awarded Grant #OT2 HD107558-01 from the Foundation for the National Institutes of Health (https://www.nih.gov/). The funder did not play any role in the study design, data collection and analysis, decision to publish, or preparation of the manuscript.

**Competing interests:** The authors have declared that no competing interests exist.

**Abbreviations:** CMC, Children with Medical Complexity; RADx-UP, Rapid Acceleration of Diagnostics-Underserved Populations; CCC, Complex Chronic Condition; RUCA, Rural-Urban Commuting Area.

## Conclusions

While rural and urban CMC caregivers reported generally high access and ease of COVID-19 testing, potentially modifiable factors exist to improve testing intention and decrease barriers, including communication regarding testing utility and timing as well as access to effective treatment response upon testing positive.

## Introduction

COVID-19 testing stands as an invaluable tool in long-term pandemic management, optimizing the health outcomes and well-being of children with medical complexity (CMC) [1]. For CMC, defined as children with complex chronic medical conditions causing severe functional limitations, consistent access to testing facilitates early detection and symptom monitoring, implementation of clinical action plans, and consideration of COVID-19–directed therapies [2,3]. However, the utility of COVID-19 testing is limited by barriers to access and perceptions surrounding use [4–8].

Early in the pandemic, CMC were three times more likely to experience severe illness and eight times more likely to be hospitalized for COVID-19 compared to the general pediatric population, while also encountering decreased access to vital health services [9,10]. For rural CMC and their caregivers, these pandemic-era challenges exacerbated the long-standing gaps in rural healthcare access and quality [11]. Compared to CMC in non-rural environments, rural CMC experience higher rates of emergency department visits and lower rates of primary care and medical specialty visits, suggesting disparities in access to care and unmet health needs [12,13]. Prior research has further demonstrated worse access to COVID-19 testing for rural populations, often stemming from an absence of accessible testing locations, financial constraints, or misinformation [4–8]. Despite the known inequities faced by children living at the intersection of rurality and medical complexity, this continues to be a crucial but under-researched domain.

Our objective was to investigate associations between rurality and COVID-19 testing access, intent, motivators, and concerns for caregivers of CMC. By better understanding rural-urban differences in COVID-19 testing perceptions, clinicians, researchers, and policymakers can effectively target modifiable factors to enhance testing uptake. Further, by using this research to investigate why urban-rural disparities exist, we can design more effective and equitable public health and policy responses to mitigate pathogen transmission in future pandemics, especially for CMC living in both urban and rural communities.

## Methods

### Study design, setting, and participants

We conducted a cross-sectional analysis of survey data collected from April 1, 2022 - June 30, 2022 at an academic medical center in the Midwestern region of the United States, as part of the National Institutes of Health's Rapid Acceleration of

Diagnostics-Underserved Populations (NIH's RADx-UP) return-to-school initiative [14]. Survey content was derived from the NIH's RADx-UP Common Data Elements library [15].

Eligible respondents were English- or Spanish-speaking primary caregivers of children aged 5–17 years with school attendance before March 2020, at least two encounters at our medical center in 2020, and at least one complex chronic condition (CCC), defined as a medical condition expected to last at least 12 months and involve several different organ systems severely enough to require specialty care and probable hospitalization at a tertiary care center [16]. All participants identified as eligible via the electronic health record were invited to participate through web and mail surveys.

As our study focused on children with medical complexity and included socially vulnerable populations, such as non-English speakers, we took measures to mitigate any risk posed to participants. Consent information and study materials were written in plain language and translated into Spanish, and participation was limited to adult parents or guardians via mailed surveys. At our institution, certified Spanish translators reviewed and translated all items, with a second translator performing back-translation and verification to ensure accuracy and consistency. Written consent was obtained by mail and the study was approved by the university's institutional review board. Results were reported following the Strengthening the Reporting of Observational Studies in Epidemiology (STROBE) reporting guidelines.

## Outcomes

The outcomes represented COVID-19 testing access, intent, motivators, and concerns. Testing access and intent variables were presented as statements (e.g., "It is easy for my child to get tested for COVID-19.") with agreement collected on a 5-point Likert scale. Responses were dichotomized as 1 = "Strongly agree" or "Agree" and 2 = "Neither agree nor disagree," "Disagree," or "Strongly disagree."

Additionally, testing motivators (e.g., "Reduce worry that my child has COVID-19") and concerns ("My child may experience discomfort from being tested") were presented as responses to the questions "How much do the following [encourage/discourage] you to get your child tested?" Agreement was indicated on a 5-point scale and dichotomized as 1 = "Very much" or "Moderately" and 2 = "Somewhat," "Slightly," or "Not at all."

## Primary exposure

Rurality was defined using participant zip code matched to the Rural-Urban Commuting Area (RUCA) codes, which accounts for measures of population density, urbanization, and daily commuting [17]. Rural residence was defined as RUCA levels 4–10 (Isolated Rural Area, Small town, and Micropolitan) and urban residence as RUCA levels 1–3 (Metropolitan) [18].

## Covariates

Covariates were selected *a priori* and included demographic and clinical variables consistent with prior research and suspected to have associations with rurality status and the study outcomes [19,20]. Demographic covariates included caregiver education level, family annual income, health insurance type (private, public and/or private, none), language spoken at home (English or language other than English), and child age (5–10, 11–13, or ≥ 14). Clinical variables included the number of CCCs (1 or ≥ 2), perceived severity of COVID-19 on child's health, and number of hospital encounters in 2020 (0 or ≥ 1). As previous studies have demonstrated disparities in healthcare access resulting from systemic and structural racism [12,21], we considered race and ethnicity as a covariate, though it was not included in the final models due to missingness and model non-convergence.

## Analysis

Summary statistics described respondents' demographic and clinical characteristics. Univariable followed by multivariable logistic regression models estimated associations between rurality and COVID-19 testing intent, access, motivators, and

concerns. Covariates with statistically significant associations were included in the multivariable models, as were those included from prior research regardless of statistical significance [19,20]. Model diagnostics were conducted to assess model fit and stability. Unadjusted and adjusted odds ratios (aOR) are reported. To investigate the potential influence of response bias, we conducted a sensitivity analysis using Chi-squared tests to assess differences between respondents and nonrespondents based on available data from the creation of the eligible cohort. Analyses were conducted in SAS software v.9.4 (SAS Institute, Cary, NC). P-values < .05 were considered statistically significant.

## Results

### Participants

Among 2,897 eligible caregivers, 1,432 returned surveys (response rate 49%). Overall, 359 (25%) were classified as rural (Table 1). Rural caregivers were significantly more likely to have lower household annual income and caregiver educational attainment, while urban caregivers were significantly more likely to have private insurance for their child and speak a language other than English.

Rural and urban children were similar regarding sex and age, but rural children were significantly more likely than urban children to have two or more complex chronic conditions (p = 0.005). Rural caregivers were also significantly more likely than urban caregivers to perceive their child as being at risk of severe health effects from COVID-19 (p = 0.03).

Respondents and non-respondents exhibited similar distributions of child age, number of CCCs, and rurality status. However, a greater proportion of respondents identified as non-Hispanic White and reported private insurance use as compared to non-respondents (S1 Table).

### Testing intent and access

In the multivariable models (Table 2), rural and urban caregivers reported similar, notably high, testing access: "I know where I can get COVID-19 testing for my child in our community" (94% and 94%, respectively) and "It is easy for my child to get tested for COVID-19" (83% and 84%, respectively). However, rural caregivers were significantly less likely to "get my child tested as often as needed" compared to urban caregivers (aOR [95% CI]: 0.53 [0.40, 0.71]).

### Testing motivators and concerns

Rural caregivers were significantly less likely than urban caregivers to select any of the given motivators as encouraging testing, e.g., to reduce worry, mitigate transmission, facilitate school attendance, or seek early treatment (Table 3). Notably, the largest effect sizes for rural caregivers were "Belief that my child was exposed to someone who has COVID-19" (0.47 [0.36, 0.62]) and "Reduce worry that my child might have COVID-19" (0.55 [0.42, 0.72]). However, testing "to let my child's school know that they are safe to attend in-person" was the most common motivator for both rural and urban caregivers (64% and 76%, respectively), followed closely by testing "to get my child treated early (if they are positive)" (63% and 71%, respectively).

Concerns with testing for COVID-19 revolved around procedure discomfort, the value or necessity of testing, and resource accessibility. Rural caregivers were significantly more likely than urban caregivers to indicate that "it will be difficult to get needed healthcare if my child has it" (2.49 [1.19, 5.18]). Overall, the primary concern with testing for rural and urban caregivers was "My child doesn't have COVID-19 symptoms so they don't need to be tested" (26% and 24%, respectively), followed by "Even if they don't have it when tested, my child can still get COVID-19 later." (15% and 9%, respectively).

## Discussion

This study is among the first to quantify COVID-19 testing perceptions among CMC caregivers, demonstrating key differences between rural and urban populations. Although most rural caregivers reported that it was easy to test their child for COVID-19, four-in-ten still indicated that they did not intend to test their child as often as needed, even amidst the

**Table 1.** Demographic and clinical characteristics of caregivers and children overall and by rurality status[1.]

| | All<br>n (%) | Rural<br>n (%) | Urban<br>n (%) | p-value[2] |
|---|---|---|---|---|
| Total, n (row %) | 1432 (100) | 359 (25) | 1073 (75) | |
| **Caregiver Characteristics** | | | | |
| Age | | | | |
| Median (IQR) | 43 (39 - 48) | 41 (37 - 46) | 43 (39 - 48) | <.0001 |
| Parent Race and Ethnicity | | | | |
| White, Non-Hispanic | 1205 (84) | 322 (90) | 883 (82) | 0.0002 |
| Hispanic | 78 (5) | 8 (2) | 70 (7) | |
| Black, Non-Hispanic | 24 (2) | 2 (0.5) | 22 (2) | |
| Other race, Non-Hispanic | 47 (3) | 4 (1) | 43 (4) | |
| Multiracial, Non-Hispanic | 14 (1) | 2 (0.5) | 12 (1) | |
| Not reported | 64 (5) | 21 (6) | 43 (4) | |
| Primary Language | | | | |
| English | 1239 (87) | 321 (90) | 918 (86) | 0.0090 |
| Other | 144 (10) | 23 (6) | 121 (11) | |
| Missing | 49 (3) | 15 (4) | 34 (3) | |
| Caregiver Highest Education | | | | |
| < 12th Grade | 33 (2) | 12 (3) | 21 (2) | <.0001 |
| GED or Some College | 496 (35) | 163 (45) | 333 (31) | |
| Bachelor's Degree | 481 (33) | 109 (30) | 372 (35) | |
| Advanced Degree | 367 (26) | 59 (17) | 308 (29) | |
| Missing | 55 (4) | 16 (5) | 39 (3) | |
| Family Annual Income | | | | |
| <$35,000 | 161 (11) | 51 (14) | 110 (10) | <.0001 |
| $35,000 - $49,999 | 100 (7) | 37 (10) | 63 (6) | |
| $50,000 - $74,999 | 188 (13) | 63 (18) | 125 (12) | |
| $75,000 - $99,999 | 204 (14) | 58 (16) | 146 (14) | |
| > $100,000 | 577 (41) | 79 (22) | 498 (46) | |
| Not Reported | 202 (14) | 71 (20) | 131 (12) | |
| **Child Characteristics** | | | | |
| Sex | | | | |
| Male | 729 (51) | 184 (51) | 545 (51) | 0.17 |
| Female | 649 (45) | 157 (44) | 492 (46) | |
| Non-binary | 3 (0) | 0 (0) | 3 (0) | |
| Missing/ Not reported | 51 (4) | 18 (5) | 33 (3) | |
| Age | | | | |
| 5 - 10 years old | 466 (32) | 127 (36) | 339 (32) | 0.45 |
| 11 - 13 years old | 332 (23) | 80 (22) | 252 (23) | |
| > 14 years old | 625 (44) | 151 (42) | 474 (44) | |
| Missing | 9 (1) | 1 (0) | 8 (1) | |
| Primary Insurance Type[3] | | | | |
| Public Insurances | 522 (36) | 173 (48) | 349 (33) | <.0001 |
| Private Insurances | 866 (61) | 169 (47) | 697 (65) | |
| Other Insurances | 44 (3) | 17 (5) | 27 (2) | |

*(Continued)*

**Table 1.** (Continued)

| | All n (%) | Rural n (%) | Urban n (%) | p-value[2] |
|---|---|---|---|---|
| Number of Complex Chronic Conditions | | | | |
| 1 | 1021 (71) | 235 (65) | 786 (73) | 0.0047 |
| 2 or more | 411 (29) | 124 (35) | 287 (27) | |
| Hospital Admissions in 2020 | | | | |
| 0 | 1177 (82) | 286 (80) | 891 (83) | 0.15 |
| 1 or more | 255 (18) | 73 (20) | 182 (17) | |
| Ever Tested Positive for COVID-19 | | | | |
| Yes | 489 (34) | 132 (37) | 357 (33) | 0.03 |
| No | 840 (59) | 188 (52) | 652 (61) | |
| Missing | 103 (7) | 39 (11) | 64 (6) | |
| Perceived likelihood of severe COVID-19 health effects[4] | | | | |
| Not at all | 480 (34) | 100 (28) | 380 (36) | 0.03 |
| A little, Somewhat | 756 (53) | 207 (58) | 549 (51) | |
| Very, Extremely | 190 (13) | 50 (14) | 140 (13) | |
| Missing | 6 (0) | 2 (0) | 4 (0) | |

[1]Rurality is categorized using the Rural-Urban Continuum Area codes, where rural = isolated rural area, small town, and micropolitan; and urban = metropolitan.

[2]Differences between rural and urban participants were assessed using T-tests, Chi-square tests, and Fisher's exact tests as appropriate.

[3]The 'Public Insurances' category includes participants with both public and private insurance. The 'Other Insurances' category includes participants who selected "Does not have health insurance", "Don't Know", or "Prefer not to answer."

[4]Respondents are asked "If your child was sick with COVID-19, how likely would their health be severely affected?"

**Table 2.** Frequencies and unadjusted/adjusted associations of COVID-19 testing access and intent with rurality status among caregivers of children with medical complexity[1].

| | Frequency | | Unadjusted | | | Adjusted[2] | | |
|---|---|---|---|---|---|---|---|---|
| | Rural n (%) | Urban n (%) | OR | 95% CI | p-value | OR | 95% CI | p-value |
| Testing Access | | | | | | | | |
| I know where I can get COVID-19 testing for my child in our community. | 336 (94) | 1009 (94) | 1.03 | (0.61, 1.73) | 0.91 | 0.97 | (0.56, 1.69) | 0.92 |
| It is easy for my child to get tested for COVID-19. | 294 (83) | 900 (84) | 0.89 | (0.64, 1.22) | 0.45 | 0.92 | (0.65, 1.30) | 0.63 |
| Testing Intent | | | | | | | | |
| I plan to get my child tested as often as needed. | 218 (61) | 805 (75) | 0.52 | (0.40, 0.67) | <.0001* | 0.53 | (0.40, 0.71) | <.0001* |

[1]Testing access and intent are collected on a 5-point Likert scale and are dichotomized by 1 = Agree, Strongly agree; 2 = Neither agree nor disagree, Disagree, Strongly disagree.

[2]Covariates include caregiver education level, family annual income, health insurance coverage type, household primary language, child age, number of complex chronic conditions, perceived severity of COVID-19, and number of hospital encounters in 2020.

*Statistically significant difference determined by logistic regression models, where statistical significance is defined as p < 0.05.

observation that rural CMC in this study may have had greater severity of illness. Motivators for testing revolved around early treatment and safe school attendance, while testing concerns hinged on the utility of testing and access to healthcare. This line of research could guide tailored public health messaging and clinical or policy interventions to encourage testing uptake and mitigate health disparities for rural CMC. These insights also signal the need to explore whether similar perceptions exist for rural populations among other transmissible illnesses, such as influenza.

**Table 3. Frequencies and unadjusted/adjusted associations of COVID-19 testing motivations and concerns with rurality status among caregivers of children with medical complexity[1].**

| | Frequency | | Unadjusted | | | Adjusted[2] | | |
|---|---|---|---|---|---|---|---|---|
| | Rural n (%) | Urban n (%) | OR | 95% CI | p-value | OR | 95% CI | p-value |
| Motivations to test | | | | | | | | |
| Reduce worry that my child might have COVID-19. | 171 (48) | 688 (65) | 0.50 | (0.39, 0.64) | <.0001* | 0.55 | (0.42, 0.72) | <.0001* |
| Belief that my child was exposed to someone who has COVID-19. | 177 (50) | 716 (67) | 0.47 | (0.37, 0.61) | <.0001* | 0.47 | (0.36, 0.62) | <.0001* |
| To know if my child is safe not to give COVID-19 to friends and family. | 202 (57) | 755 (71) | 0.53 | (0.41, 0.68) | <.0001* | 0.56 | (0.43, 0.74) | <.0001* |
| To know if my child is safe not to give COVID-19 to anyone they are around. | 204 (58) | 759 (71) | 0.55 | (0.43, 0.70) | <.0001* | 0.56 | (0.43, 0.74) | <.0001* |
| To let my child's school know that they are safe to attend in-person. | 227 (64) | 801 (76) | 0.57 | (0.44, 0.74) | <.0001* | 0.59 | (0.44, 0.78) | 0.0002* |
| To get my child treated early (if they are positive). | 226 (63) | 753 (71) | 0.72 | (0.56, 0.92) | 0.0096* | 0.72 | (0.55, 0.95) | 0.02* |
| Concerns with testing | | | | | | | | |
| My child may experience discomfort from being tested. | 29 (8) | 66 (6) | 1.33 | (0.85, 2.10) | 0.22 | 0.98 | (0.59, 1.61) | 0.93 |
| Even if they don't have it when tested, my child can still get COVID-19 later. | 53 (15) | 93 (9) | 1.81 | (1.26, 2.60) | 0.0013* | 1.35 | (0.90, 2.03) | 0.15 |
| My child doesn't have COVID-19 symptoms so they don't need to be tested. | 94 (26) | 258 (24) | 1.11 | (0.85, 1.46) | 0.45 | 0.92 | (0.68, 1.24) | 0.58 |
| If my child is positive, officials will need to contact the people they've been in contact with. | 50 (14) | 94 (9) | 1.67 | (1.16, 2.41) | 0.0062* | 1.43 | (0.94, 2.18) | 0.10 |
| I don't want to know if my child has it. | 10 (3) | 26 (2) | 1.15 | (0.55, 2.40) | 0.72 | 1.18 | (0.49, 2.81) | 0.71 |
| There is not much they can do for my child if they have it. | 44 (12) | 71 (7) | 1.95 | (1.31, 2.90) | 0.0010* | 1.50 | (0.96, 2.35) | 0.07 |
| It will be difficult to get needed healthcare if my child has it. | 19 (5) | 19 (2) | 3.07 | (1.61, 5.87) | 0.0007* | 2.49 | (1.19, 5.18) | 0.02* |

[1]Testing motivations and concerns ask, "How much do the following [encourage/ discourage] you to get your child tested?" and are dichotomized by 1 = Moderately, Very much; 2 = Not at all, Slightly, Somewhat.

[2]Covariates include caregiver education level, family annual income, health insurance coverage type, household primary language, child age, number of complex chronic conditions, perceived severity of COVID-19, and number of hospital encounters in 2020.

*Statistically significant difference determined by logistic regression models, where statistical significance is defined as p < 0.05.

In contrast to prior studies underscoring the barriers rural populations face to COVID-19 testing access, e.g., transportation, cost, or misinformation [5–7], rural and urban caregivers in our study reported similarly high access to COVID-19 testing. We suspect that national and statewide testing programs, which included options to receive free in-home test kits by mail [22], overcame some challenges faced by these communities early in the pandemic. Some CMC caregivers in this sample may have been enrolled in a care coordination program at our institution, which provides interdisciplinary care and care coordination that can translate to better service access [23,24]. In addition, CMC caregivers often have extensive experience with healthcare navigation [25], and rural caregivers of CMC may be more skilled at obtaining COVID-19 tests when needed compared to rural caregivers of children without medical complexity.

It remains notable, however, that one-in-five caregivers, both rural and urban, still indicate challenges testing their child for COVID-19. In adult populations, Collie-Akers et al. similarly recognized that testing access barriers exist for both rural and urban populations, though challenges had distinct origins. For example, transportation constraints for rural populations center on far distances, while, for urban populations, it centers on the reliance of public transportation even for relatively short distances [4,8]. Qualitative research recognizing the complex interaction of barriers and facilitators for CMC caregivers in urban and rural settings could identify targeted or potentially overlapping strategies that improve access to testing for both populations.

Our observation that rural caregivers were significantly less likely than urban caregivers to test their child as often as needed despite relatively high access to COVID-19 testing aligns with existing research. A lack of clarity around testing

guidelines is a commonly cited barrier [4–8]. In our sample, a prominent concern for rural caregivers focused on the utility and timing of testing. As CMC are at a higher risk of COVID-19 infection and often have multiple points of contact between school, home nursing, and medical providers, the opportune times to test may be less obvious [26]. To address this challenge, public health messaging could leverage CMC research and caregiver perspectives to encourage testing at high-impact time frames, emphasizing the value of early detection and symptom monitoring, implementation of clinical action plans, and consideration of COVID-19–directed therapies. Coller et al. highlighted the need to incorporate contextual factors in messaging at these pivotal moments (e.g., new school year, increased virulence) to sustain testing enthusiasm in communities [2].

Even though rural caregivers had lower testing intentions, our data did not suggest that they were disregarding COVID-19. For example, almost all reported wanting to know if their child had COVID-19, and rural caregivers were more likely than urban caregivers to state that their child's health would be severely affected by COVID-19. This raises the question of why rural populations would have lower testing intentions. Our data suggest that an important explanatory variable may be definitive treatment, namely, rural caregivers were more doubtful about their child being able to receive the needed healthcare if diagnosed with COVID-19. In fact, prior research has exemplified barriers to treatment for rural CMC, including challenges accessing pediatricians, specialty providers, and appropriate, informed emergency care in local hospitals [11,27,28]. The simultaneous interaction of financial constraints, navigation of health systems and insurance, and constant decision-making compound these structural barriers [29]. An important clinical and public health implication is that, even with adequate testing access and motivation, uptake may not occur if families do not perceive access to effective responses after a positive test. Therefore, future pandemic planning must recognize that accounting for urban-rural disparities in testing access may be necessary but insufficient. Policies and public health responses to pandemics must also account for distinct motivators and barriers to definitive treatment faced by rural communities.

An important future research direction is determining whether these findings translate to other respiratory illnesses, such as influenza, RSV, or other novel pathogens. This would be imperative in planning responses to future pandemics by guiding the creation of tailored interventions, efficient resource allocation, effective community engagement, and informed policy development [2,30–32]. Further, this line of research reinforces existing evidence documenting health equity disparities within the pandemic response for CMC and their families, underscoring the importance of improving healthcare access and quality to ensure healthy lives and promote well-being for all at all ages [33,34]. As respiratory concerns are among the leading causes of hospital admission for CMC [35], healthcare providers and public health leaders must recognize the unique challenges facing the rural CMC population. Proactive measures should be taken to develop public health strategies that are tailored to respiratory illness testing motivators and designed to mitigate healthcare access barriers.

Several limitations of this study must be considered. Rurality data was obtained using 5-digit postal zip codes matched to RUCA codes. The use of zip code tabulation areas or federal information processing system codes may have improved precision [36]. As this is a single-center study at a Midwestern academic medical center in the USA, results may not be generalizable to other regions or populations. Despite efforts to mitigate selection bias through diverse recruitment methods and survey design, it remains a limitation that may also affect the generalizability of our findings. Survey questions were standardized across all RADx-UP programs and these questions did not distinguish between perceptions of COVID-19 antigen testing or polymerase chain reaction testing. Due to the diversity of over 30 reported household languages, we reported the most common languages (English and Spanish) and aggregated the remaining for analysis, as sample sizes were insufficient for detailed investigation. Other important motivators and concerns experienced by rural populations may not have been measured in our study.

Despite these limitations, our study revealed important differences in COVID-19 testing perceptions among rural and urban CMC caregivers. This study highlights both the need and value of more research with rural CMC. While rural and urban caregivers reported generally high access and ease of COVID-19 testing, opportunity remains to improve testing intention for rural caregivers. Potentially modifiable factors include communication regarding testing access and timing

as well as access to effective treatment response upon testing positive. As multiple respiratory illness epidemics persist, collaborative efforts among healthcare providers, policymakers, and public health leaders to address these challenges are vital to optimize health outcomes for rural CMC populations.

## Supporting information

**S1 Table. Caregiver and child demographic characteristics by response status, n = 3080.**
(DOCX)

## Author contributions

**Conceptualization:** Kristina Devi Singh-Verdeflor, Michelle M. Kelly, Gregory P. DeMuri, Gemma Warner, Sabrina M. Butteris, Mary L. Ehlenbach, Barbara Katz, Joseph A. McBride, Shawn Koval, Ryan J. Coller.

**Formal analysis:** Kristina Devi Singh-Verdeflor, Ryan J. Coller.

**Funding acquisition:** Ryan J. Coller.

**Investigation:** Michelle M. Kelly, Gregory P. DeMuri, Gemma Warner, Sabrina M. Butteris, Mary L. Ehlenbach, Barbara Katz, Joseph A. McBride, Shawn Koval, Ryan J. Coller.

**Methodology:** Kristina Devi Singh-Verdeflor, Michelle M. Kelly, Gregory P. DeMuri, Gemma Warner, Sabrina M. Butteris, Mary L. Ehlenbach, Barbara Katz, Joseph A. McBride, Shawn Koval, Ryan J. Coller.

**Supervision:** Ryan J. Coller.

**Writing – original draft:** Kristina Devi Singh-Verdeflor.

**Writing – review & editing:** Michelle M. Kelly, Gregory P. DeMuri, Gemma Warner, Sabrina M. Butteris, Mary L. Ehlenbach, Barbara Katz, Joseph A. McBride, Shawn Koval, Ryan J. Coller.

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
