## [Decision Letter · Decision Letter 0]

19 Dec 2024

PONE-D-24-43670Differences in COVID-19 testing perceptions from caregivers of children with medical complexity by ruralityPLOS ONE

Dear Dr. Singh-Verdeflor,

Thank you for submitting your manuscript to PLOS ONE. After careful consideration, we feel that it has merit but does not fully meet PLOS ONE’s publication criteria as it currently stands. Therefore, we invite you to submit a revised version of the manuscript that addresses the points raised during the review process.

We look forward to receiving your revised manuscript.

Kind regards,

Abdulaziz T. Bako, MBBS; MPH; PhD

Academic Editor

PLOS ONE

**Journal Requirements:**

2. In the online submission form, you indicated that The datasets used and/or analyzed during the current study are available from the corresponding author on reasonable request.

Reviewers' comments:

Reviewer's Responses to Questions

**Comments to the Author**

1. Is the manuscript technically sound, and do the data support the conclusions?

Reviewer #1: Yes

Reviewer #2: Yes

2. Has the statistical analysis been performed appropriately and rigorously? 

Reviewer #1: Yes

Reviewer #2: Yes

3. Have the authors made all data underlying the findings in their manuscript fully available?

Reviewer #1: Yes

Reviewer #2: Yes

4. Is the manuscript presented in an intelligible fashion and written in standard English?

Reviewer #1: Yes

Reviewer #2: No

5. Review Comments to the Author

**Reviewer #1: ** Thank you for your time and efforts conducting the study. Please consider the following:

1- Improve the abstract section and add a background section

2- Detail financial support received to fund the study

3- Relate the study to global commitment to universal health coverage and SDG3

4- Shed light on the impact of this study on future pandemic planning

5- Detail the rationale of the study

6- Explain the validation process for translated questionnaire and training of interviewers

7- Explain ethical consideration that had been accounted for vulnerable population

8- What are other languages in Table 1

**Reviewer #2: ** In this paper, the authors conduct a cross-sectional survey of English- and Spanish-speaking primary caregivers of children with chronic conditions at an academic medical center in the Midwest.

The paper reports that, compared to urban caregivers, rural caregivers had significantly less testing intent and were significantly more likely to indicate difficulties in gaining access to needed healthcare. The authors conclude that potentially modifiable factors can improve perceived barriers associated with COVID-19 testing.

Overall, the paper presents a well-developed approach to investigating associations between rurality (and other explanatory variables) and COVID-related outcome variables in rural-urban areas. The research questions explored in this study – if repeated in a number of robust studies – can be of significance for the implementation of public health interventions in times of pandemics and other public health emergencies.

With that in mind, this reviewer has the following to remark:

Introduction

1.1 The introduction section begins with this sentence: “COVID-19 testing is an invaluable tool in long-term pandemic management.[1]”

This way of in-text citation was used throughout the article. It must be noted that references are usually placed before punctuation (e.g., periods, commas, semicolons).

Therefore, it would be prudent to look at reference placement in the entire manuscript and make any necessary adjustments.

1.2 This section is in need of a more compelling rationale that carries greater weight than the one presented here.

Methods

2.1 In the methods section, the authors write, “Survey content was derived from NIH’s RADx-UP Common Data Elements library[15]”

There should be a space between the reference and the text. And there should be a period after the reference.

Please refer to the PLOS style guidelines and format your manuscript accordingly.

2.2 Additionally, transitions between paragraphs can be improved to make the reading smoother and more fluid.

2.3 Another important issue is the sampling technique. The authors have not mentioned what sampling technique they used. It is advisable to specify it for reasons of clarity and because of its impacts on the credibility of their findings.

Discussion

3.1 In this type of studies, the participants (i.e., those who respond to the survey) are more likely to be willing to take part in the study because they may be interested in the topic for a variety of reasons, or simply because they have extra time on their hands.

I wonder if the authors have taken this aspect into consideration.

3.2 The authors state that “almost all reported wanting to know if their child had COVID-19, and rural caregivers were more likely than urban caregivers to state that their child’s health would be severely affected from COVID-19. This raises the question why rural populations would have lower testing intentions.”

Could this observation be partly related to the nature of this study and the point raised in 3.1?

I hope this review is helpful and wish the authors the very best with their research!

6. PLOS authors have the option to publish the peer review history of their article (what does this mean? ). If published, this will include your full peer review and any attached files.

**Do you want your identity to be public for this peer review?** For information about this choice, including consent withdrawal, please see our Privacy Policy .

Reviewer #1: **Yes: ** WEAM BANJAR

Reviewer #2: No

---

## [Author Response · Author response to Decision Letter 1]

14 Jan 2025

Thank you for the opportunity to revise and resubmit our manuscript. In the Data Availability section, we have specified the direct location for data accessibility within the NIH RADx Data Hub.

Responses to reviewers have been compiled for each comment in the "Response to Reviewers" document.

---

## [Decision Letter · Decision Letter 1]

16 Feb 2025

PONE-D-24-43670R1Differences in COVID-19 testing perceptions from caregivers of children with medical complexity by ruralityPLOS ONE

Dear Dr. Singh-Verdeflor,

Thank you for submitting your manuscript to PLOS ONE. After careful consideration, we feel that it has merit but does not fully meet PLOS ONE’s publication criteria as it currently stands. Therefore, we invite you to submit a revised version of the manuscript that addresses the points raised during the review process.

1. Please address the reviewer's concern by shortening the abstract section 2. Replace "an academic medical center in the Midwest" in the methods section of the abstract with "an academic medical center in the Midwestern region of the United States" or other expressions indicating that this study was conducted in the United States. 3. Please thoroughly copyedit the manuscript to identify and address typos ("decrease" was written as "decease" in the conclusion section of the abstract.

We look forward to receiving your revised manuscript.

Kind regards,

Abdulaziz T. Bako, MBBS; MPH; PhD

Academic Editor

PLOS ONE

Journal Requirements:

Reviewers' comments:

Reviewer's Responses to Questions

**Comments to the Author**

1. If the authors have adequately addressed your comments raised in a previous round of review and you feel that this manuscript is now acceptable for publication, you may indicate that here to bypass the “Comments to the Author” section, enter your conflict of interest statement in the “Confidential to Editor” section, and submit your "Accept" recommendation.

Reviewer #1: All comments have been addressed

Reviewer #2: All comments have been addressed

2. Is the manuscript technically sound, and do the data support the conclusions?

Reviewer #1: Yes

Reviewer #2: (No Response)

3. Has the statistical analysis been performed appropriately and rigorously? 

Reviewer #1: Yes

Reviewer #2: (No Response)

4. Have the authors made all data underlying the findings in their manuscript fully available?

Reviewer #1: Yes

Reviewer #2: (No Response)

5. Is the manuscript presented in an intelligible fashion and written in standard English?

Reviewer #1: Yes

Reviewer #2: (No Response)

6. Review Comments to the Author

Reviewer #1: Thank you for addressing all comments. However, it might be relevant to shorten the abstract section.

Reviewer #2: (No Response)

7. PLOS authors have the option to publish the peer review history of their article (what does this mean? ). If published, this will include your full peer review and any attached files.

**Do you want your identity to be public for this peer review?** For information about this choice, including consent withdrawal, please see our Privacy Policy .

Reviewer #1: **Yes: ** WEAM BANJAR

Reviewer #2: No

---

## [Author Response · Author response to Decision Letter 2]

4 Mar 2025

Thank you for the opportunity to revise and resubmit this manuscript. All comments have been addressed in the response to reviewers document.

---

## [Decision Letter · Decision Letter 2]

13 Apr 2025

Differences in COVID-19 testing perceptions among caregivers of children with medical complexity by rurality

PONE-D-24-43670R2

Dear Dr. Singh-Verdeflor,

We’re pleased to inform you that your manuscript has been judged scientifically suitable for publication and will be formally accepted for publication once it meets all outstanding technical requirements.

Kind regards,

Abdulaziz T. Bako, MBBS; MPH; PhD

Academic Editor

PLOS ONE

Additional Editor Comments (optional):

Reviewers' comments:

Reviewer's Responses to Questions

**Comments to the Author**

1. If the authors have adequately addressed your comments raised in a previous round of review and you feel that this manuscript is now acceptable for publication, you may indicate that here to bypass the “Comments to the Author” section, enter your conflict of interest statement in the “Confidential to Editor” section, and submit your "Accept" recommendation.

Reviewer #1: All comments have been addressed

2. Is the manuscript technically sound, and do the data support the conclusions?

Reviewer #1: Yes

3. Has the statistical analysis been performed appropriately and rigorously? 

Reviewer #1: Yes

4. Have the authors made all data underlying the findings in their manuscript fully available?

Reviewer #1: Yes

5. Is the manuscript presented in an intelligible fashion and written in standard English?

Reviewer #1: Yes

6. Review Comments to the Author

Reviewer #1: Thank you for taking the time to update the manuscript to this version. We appreciate considering reviewers comments.

7. PLOS authors have the option to publish the peer review history of their article (what does this mean? ). If published, this will include your full peer review and any attached files.

**Do you want your identity to be public for this peer review?** For information about this choice, including consent withdrawal, please see our Privacy Policy .

Reviewer #1: **Yes: ** WEAM BANJAR

---

## [Editor Report · Acceptance letter]

PONE-D-24-43670R2

PLOS ONE

Dear Dr. Singh-Verdeflor,

I'm pleased to inform you that your manuscript has been deemed suitable for publication in PLOS ONE. Congratulations! Your manuscript is now being handed over to our production team.

Kind regards,

on behalf of

Dr. Abdulaziz T. Bako

Academic Editor

PLOS ONE